# A Complexity-Entropy Based Approach for the Detection of Fish Choruses

**DOI:** 10.3390/e21100977

**Published:** 2019-10-06

**Authors:** Shashidhar Siddagangaiah, Chi-Fang Chen, Wei-Chun Hu, Nadia Pieretti

**Affiliations:** 1Underwater Acoustic Laboratory, Department of Engineering Sciences and Ocean Engineering, National Taiwan University, Taipei 106, Taiwan; william_hu@outlook.com; 2Department of Life and Environmental Science, Polytechnic University of Marche, 60131 Ancona, Italy; nadia.pieretti@gmail.com

**Keywords:** Eastern Taiwan Strait, ecoacoustics, acoustic indices, marine environments, fish choruses, passive acoustic monitoring

## Abstract

Automated acoustic indices to infer biological sounds from marine recordings have produced mixed levels of success. The use of such indices in complex marine environments, dominated by several anthropogenic and geophonic sources, have yet to be understood fully. In this study, we introduce a noise resilient method based on complexity-entropy (hereafter named C-H) for the detection of biophonic sounds originating from fish choruses. The C-H method was tested on data collected in Changhua and Miaoli (Taiwan) during the spring in both 2016 and 2017. Miaoli was exposed to continual shipping activity, which led to an increase of ~10 dB in low frequency ambient noise levels (5–500 Hz). The acoustic dataset was successively analyzed via the acoustic complexity index, the acoustic diversity index and the bioacoustic index. The C-H method was found to be strongly correlated with fish chorusing (Pearson correlation: r_H_ < −0.9; r_C_ > 0.89), and robust to noise originating from shipping activity or natural sources, such as wind and tides (r_H_ and r_C_ were between 0.22 and −0.19). Other indices produced lower or null correlations with fish chorusing due to missed identification of the choruses or sensitivity to other sound sources. In contrast to most acoustic indices, the C-H method does not require a prior setting of frequency and amplitude thresholds, and is therefore, more user friendly to untrained technicians. We conclude that the use of the C-H method has potential implications in the efficient detection of fish choruses for management or conservation purposes and could help with overcoming the limitations of acoustic indices in noisy marine environments.

## 1. Introduction

Underwater sound exists throughout the world’s oceans. Recently, the issue of an increasing contribution from human activities into the marine environment has received increased focus from scientists and international institutions [1,2,3]. Augmented noise input from human activities in the ocean originates mainly from continual shipping, offshore and coastal construction and industrialization, and such noise has been found to produce severe negative impacts on marine species [4,5,6,7,8,9,10]. To quantify the abundance, behavior and diversity of species in relation to noise effects, one must develop efficient and reliable monitoring tools. Recently, passive acoustic monitoring (PAM) has been used for the long-term monitoring of marine biodiversity and as a proxy for population health [11,12,13,14]. The implementation of PAM has resulted in the collection of vast datasets [15]. Nevertheless, extracting significant details regarding marine communities or specific species remains a challenging task [16,17].

Acoustic Indices (AIs) are utilized to rapidly quantify the biophony and anthropophony from large datasets collected by PAM [18,19]. AIs allow us to represent an acoustic recording as a single value and analyze biophony and anthropophony at different ecological levels. AIs depend on the vocalization of the species and the characteristics of the recordings, including the diversity, density and complexity of the sounds in the recordings coupled with the ratio of the sounds throughout frequencies [20]. An ideal AI should be able to measure the variation of acoustic energy over temporal and spatial scales in accordance with the vocalizing species in the acoustic recordings. An increased variability in the soundscape due to enhanced biophony will result in higher values of AI. Therefore, AI allows the inferring of important information about biodiversity and abundances of species in a given environment.

AIs were initially created for studying terrestrial ecosystems [20] and have been successively applied in marine environments [21,22,23,24,25]. Nevertheless, AIs in marine studies have produced mixed results, particularly when anthropogenic noise was present [26,27]. Loud noise (i.e., air guns) mask the weaker biological signals [24], whereas continuous noise from vessels covers the frequency band used by fish and impedes the selection of a usable part of the spectrogram [21]. There are over 30 AIs in use in terrestrial ecoacoustics [20,28]. Some of the AIs, such as the acoustic complexity index (ACI) [29], acoustic diversity index (ADI) [30,31] and bioacoustic index (BI) [32], have been utilized to explore trends in biophony produced by various species of mammals, fish and crustaceans [26,27,33]. Harris et al. employed ACI and entropy indices for quantifying coral reef species’ diversity and proposed a criterion for ideal use of AIs in a marine environment [34]. Subsequently, several studies have utilized AIs for studying marine soundscapes and studying temporal and nocturnal chorusing patterns in marine fauna [21,26,33,35,36].

Generally, AIs work on the principle of evaluating the sound energy or the heterogeneity/complexity of sounds in a specified frequency interval or time duration [28]. Sound energy will result in higher values when biophony or noise occurs in the recordings. The Shannon entropy is utilized in AIs for addressing the heterogeneity of sounds, but this approach poses a number of drawbacks [37]. First, Shannon’s measure neglects temporal relationships occurring in-between the values of acoustic time series [38], thereby overlooking the structure and possible temporal patterns existing in the acoustic recordings. Second, its applicability depends on particular features of the data, such as stationarity, time series length, frequency range and degree of noise contamination. Third, it is often suitable for linear systems, making it unsuitable for ambient ocean noise, which is characterized by complex nonlinear fluctuations and is non-stationary [39].

To address this issue, Bandt and Pompe introduced permutation entropy obtained by Shannon entropy, evaluated via the ordinal scheme to calculate the probability distribution associated with the time series [40]. The combination of permutation entropy with statistical complexity introduced by Rosso et al. [41] is referred to as the complexity-entropy method (C-H method). This method has been utilized in quantifying both degrees of randomness and temporal correlation structures in time series with complex fluctuations, existing in diverse fields, such as finance, climate, medicine, geophysics and acoustics [42,43,44].

In this study, we propose the C-H method as a new tool for the automated detection of fish choruses and test its validity on recordings taken at Changhua (A1) and Miaoli (N1) located at the Eastern Taiwan Strait (ETS) (see Figure 1). As means of comparison, the ACI [22], ADI [23,24] and BI [25] were also used to process the files. We selected these three AIs because they were built to be robust to anthropogenic noise while characterizing the level of biophony in the acoustic recordings. The study in [16] also chose those three AIs for assessing biases in acoustically complex urban habitats dominated by anthropogenic noise. We hypothesize that the C-H method can quantify the change in randomness and correlation structure occurring in the time series [41] and enable the detection of fish chorusing within the fluctuating ambient ocean noise [43]. The capability of the C-H method of assessing changes in acoustic periodicities is proposed herein as a novel approach that differs from currently utilized AIs, such as ACI, ADI and BI.

## 2. Material and Methods

### 2.1. Data Measurements

Passive acoustic recorders were deployed in two regions, Changhua (A1) and Miaoli (N1), located approximately 4.5 km off the Eastern Taiwan Strait (ETS) (Figure 1). A1 and N1 are 90 km apart located in the Taiwan Strait and are composed of sandy bottoms and hard sedimentary rocks at 20 to 25 m of depth. Monitoring duration, depth of recorders and geographical coordinates of both recording stations are reported in Table 1. The recordings were performed utilizing wildlife acoustics SM3M (sensitivity −164.5 dB re 1 µPa; flat frequency response (±0.1 dB) between 0.1 to 10 kHz). Acoustic files of 60 min were recorded continuously with a sampling frequency of 48 kHz, wav format, encompassing ~3840 h and ~1.3 TB of acoustic recordings.

Diverse sources of sounds affect the composition of the acoustic environment of both areas. The northeast monsoon in winter and the southwest monsoon in summer are the primary factors affecting the wind and current variations around Taiwan [45]. Every year the ETS experiences several tropical storms, accompanied by strong wind speeds. These extreme meteorological conditions cause the sea’s tidal level rise and flooding to occur, thus triggering sediment transport and noise due to the collision of sediment grains. Noise from maritime activities is also present at the study sites. In particular, Long Feng fishing harbor is situated 5 km from N1, causing dense vessel activity in that area. Fishes at both A1 and N1 are the major contributors to the biological sounds. Sciaenidae species are found abundantly at the ETS and are known to produce choruses [46,47]. These species are commonly addressed as croakers or drums. The chorusing is carried-out for attracting mates during spawning, communication and social cohesion. Indo-pacific humpback dolphins and bottlenose dolphins are also known to inhabit the ETS [48,49].

### 2.2. The Complexity-Entropy Method

The information content of a system is typically computed from a probability distribution *P* describing the distribution of some observable quantity. An information measure can primarily be viewed as a quantity that characterizes that given probability distribution. Shannon entropy is utilized as a measure to quantify the information content of a system. Given any arbitrary probability distribution $P\;=\;\left\{p_{i}:i\;=\;1,\ldots ,M\right\}$, the Shannon’s logarithmic information, $S\left[P\right]\;=\;-\overset{M}{\underset{i\;=\;1}{\sum }}p_{i}\ln p_{i}$, is regarded as the measure of the uncertainty associated with the physical process described by *P*. When *S*[*P*] = 0, it is possible to predict with complete certainty, which of the possible outcomes *i* (the probabilities of which are given by *p_i_*) will actually take place. In this instance, the underlying process described by the probability distribution is the maximum compared to a uniform distribution [50].

Nevertheless, an entropy measure cannot quantify the degree of structure or patterns present in a process [51]. Moreover, it was shown that measures of complexity can quantify the degree of correlation structures. This kind of information is not discriminated by randomness measures. The opposite extremes are perfect order and maximal randomness (for example, a periodic sequence and white noise). The statistical complexity measure (C) introduced by Lamberti et al. [52] can provide a wide range of possible degrees of correlation structure between these extremes. Statistical complexity can detect essential correlation structures and provides important additional information regarding the characteristics of the underlying probability distribution, which entropy cannot detect. Statistical complexity (C) is a function of *P* and is defined through the product [53]
(1)$$C\left[P\right]\;=\;Q_{j}\left[P,P_{e}\right]H_{s}\left[P\right]$$
of the normalized Shannon entropy
(2)$$H_{s}\left[P\right]\;=\;S\left[P\right]/S_{max}$$
with $S_{max}\;=\;S\left[P_{e}\right]\;=\;\ln M$
$\left(0\leq H_{s}\left[P\right]\leq 1\right)$ and $P_{e}\;=\;\left\{1/M,\ldots ,\;1/M\right\}$ as the uniform distribution, and the disequilibrium $Q_{J}$ being defined as, $Q_{J}\left[P,P_{e}\right]\;=\;Q_{0}J\left[P,P_{e}\right]$ with $J\left[P,P_{e}\right]\;=\;\left\{S\left[P+P_{e}/2\right]-S\left[P\right]/2-S\left[P_{e}\right]/2\right\}$ as the Jensen–Shannon divergence, and $Q_{0}$, a normalization constant being the inverse of the maximum possible value of $J\left[P,P_{e}\right]$. This value is obtained when one of the components of *P*, say *p_m_* = 1, and the remaining components are equal to zero. The Jensen–Shannon divergence quantifies the difference between the probability distribution *P* and *P_e_* [54]. C is not a trivial function of the entropy because it depends on two different probabilities’ distributions: the one associated with the system under analysis, *P*, and the uniform distribution, *P_e_*. Furthermore, for each given value of the normalized permutation entropy $H_{s}\left[P\right]\in \left[0,1\right]$, there is a range of possible values of complexity, C_min_ ≤ C ≤ C_max_. The usual procedure for evaluating the bounds C_min_ and C_max_ is given in Martin et al., 2006 [55]. Thus, it is clear that important additional information related to the correlational structure between the components of the physical system is provided by evaluating C.

To calculate *H_S_* and C, an associated probability distribution should be constructed. The probability distribution is evaluated by using the method introduced by Bandt and Pompe [40]. This method considers the time causality of the system dynamics, performed by employing a suitable partition based on ordinal patterns obtained by comparing neighboring values of the original series. For a given time series ${\left\{x\left(t\right)\right\}}_{t\;=\;1}^{N}$, of an embedding dimension *d* > 1, and an embedding delay time *τ*, the ordinal pattern of order *d* generated by
(3)$$s\mapsto \left(x_{s-\left(d-1\right)\tau },x_{s-\left(d-2\right)\tau },\ldots ,x_{s-\tau },x_{s}\right)$$
has to be considered. To each time *s*, we assign a *d*-dimensional vector that results from the evaluation of the time series at times *s* − (*d* − 1) *τ*, …, *s* − *τ*, *s*. Clearly, the higher the value of *d*, the more the information about the past that is incorporated into the ensuing vectors. By the ordinal pattern of order *d* related to time *s*, we mean the permutation $\pi \;=\;\left(r_{0},r_{1},\ldots ,r_{D-1}\right)$ of (0, 1, …, *d* − 1) defined by

(4)$$x_{s-r_{0}\tau }\geq x_{s-r_{1}\tau }\geq \ldots \geq x_{s-r_{d-2}\tau }\geq x_{s-r_{d-1}\tau }$$

In this manner, the vector defined by Equation (3) is converted into a unique symbol π. For instance, if we assume a time series {7, 3, 4, 5, 2, 9, …} with embedding dimension *d* = 4 and embedding delay *τ* = 1, the state space is divided into 4! partitions, and 24 mutually exclusive permutation symbols are considered. The first four-dimensional vector is (7, 3, 4, 5). According to Equation (3), this vector corresponds to $\left(x_{s-3},x_{s-2},x_{s-1},x_{s}\right)$, Following Equation (4), it yields $x_{s-3}\geq x_{s}\geq x_{s-1}\geq x_{s-2}$. Then, the ordinal pattern which satisfies Equation (4) will be (3, 0, 1, 2). The second four-dimensional vector is (3, 4, 5, 2) and (1, 2, 3, 0) will be its associated permutation, and so on. For all the *D*! possible permutations *π_i_* of order *d*, their associated relative frequencies can be naturally computed by the number of times this particular order sequence is found in the time series divided by the total number of sequences. Thus, an ordinal pattern probability distribution $P\;=\;\left\{p\left(\pi_{i}\right),i\;=\;1,\ldots ,\;d!\right\}$ is obtained from the time series. Determining the embedding dimension *d* and the time delay *τ* are necessary to evaluate the appropriate probability distribution, particularly for *d*, which determines the number of accessible states given by *d*!. Moreover, it was established that the length *N* of the time series must satisfy the condition N ≫
*d*! in order to obtain reliable statistics [56]. For practical purposes, Bandt and Pompe suggested *d* = 3, …, 7 and embedding delay *τ* = 1 [40].

The above discussed, the normalized Shannon entropy *H_S_*—Equation (2), and the statistical complexity C—Equation (1), are calculated using the permutation probability distribution $P\;=\;\left\{p\left(\pi_{i}\right),i\;=\;1,\ldots ,\;d!\right\}$. Defined in this manner, these quantifiers are typically referred to as permutation entropy (H) and permutation statistical complexity (C) [57]. The H and C produce complementary information about two distinct properties of a data set. Permutation entropy quantifies the degree of disorder inherent in a process, where more predictable signals, which demonstrate a tendency to repeat just a few fluctuation patterns, have lower permutation entropy than less predictable signals (which tend to exhibit all the possible fluctuation patterns). For a given entropy value, the statistical complexity quantifies the degree to which there exists privileged fluctuation patterns among those patterns accessible to the system. By calculating these quantities for a given time series, both randomness and the degree of correlational structures in the fluctuations of the system are evaluated simultaneously [41]. Complexity and entropy were evaluated using the R package ‘statcomp’ (Version 0.0.1.1000) [58] using the command global-complexity(), for embedding dimension *d* = 6, and to evaluate the C_max_ and C_min_, command limit_curves(*d*, fun = ‘min/max’) is utilized (R code is provided in Appendix A).

### 2.3. Computation of Acoustic Indices

In this study, we utilized three AIs, namely ACI [29], ADI [30,31] and BI [32], to explore the detection capability of fish chorusing and to verify the robustness of these indices to anthropogenic noise. ACI measures the sound intensity in different, short duration frequency bins, yielding high ACI values for biophony from fish calls and low values in the presence of constant sounds originating from anthropophony from shipping and ambient noise. ADI calculates the Shannon’s diversity index to evaluate spectrum complexity along width-prefixed frequency bins and evaluates the degree of variation of energy along frequencies of a selected portion of the spectrogram, thus yielding high ADI values when biophony occurs. BI is the area under the power spectrum curve across the specified range of frequencies occupied by the biophony and is used to describe the proportion of this acoustic space that occupies a recording. These acoustic indices were calculated using the R (version 3.4.1) [59] package ‘soundecology’ (version 1.3.2) [60]. ACI and BI were applied to the frequency range 10–3500 Hz with an Fast Fourier Transform (FFT) of 2048 points (frequency resolution 39.06 Hz; temporal resolution 0.026 s) and a cluster size of 60 s for ACI computations. For calculating ADI, the maximum frequency was set to 3500 Hz, fixing a resolution of 200 Hz and a threshold of −50 dB.

### 2.4. Data Analysis and Statistics

Spectrograms and sound pressure levels (SPLs) were computed using PAMGuide [61] with an FFT size of 1024 points and a 1-s time segment averaged to 60-s resolution via the Welch method. Mean (RMS), median and 95th percentile SPLs were calculated for the frequency bands 50 to 200 Hz and 500 to 2500 Hz, which were chosen as reference bands for quantifying low frequency anthropogenic noise and fish chorusing, respectively. Most of the fish chorusing was observed in the frequency band 500 to 2500 Hz, hence this was chosen as a reference band for chorusing activity and to avoid the effect of anthropophony occurring at other frequencies. To explore the circadian pattern of the fish chorusing, the hourly RMS (mean), SPL (500–2500 Hz) and standard deviation (SD) were evaluated at A1 and N1 for both years.

To assess the fish chorus detection of AIs and C-H, we firstly compared hourly mean SPL_500–2500Hz_ with hourly AIs and C-H of two recording days (h24) at A1, precisely, on 4 and 26 April 2016. Successively, we analyzed four days characterized by different levels of fish chorusing and noise: (a) intense dawn and dusk chorusing with sediment-generated noise; (b) mild dawn and dusk chorusing with vessel and sediment-generated noise; (c) mild dawn chorus with noise from 18 vessels (observed during 24 h); (d) absence of chorusing with continual, sediment-generated noise. Hourly ACI, ADI, BI and C-H were used for the analyses. A complexity-entropy plane H × C was built to show the variation of the complexity as a function of its entropy. This plane was utilized to study changes in the dynamics (randomness and correlation structure) occurring in the acoustic recordings.

Successively, AIs and C-H methods were extended to the complete monitoring at A1 and N1, during the spring periods of the years 2016 and 2017 (Table 2), with the aim of comparing their ability to infer fish choruses in long-term analysis. All indices and SPL were normalized by subtracting the minimum in the series, then dividing by the new maximum of that series. A Pearson correlation was applied to verify the relationship between indices and SPL_500–2500Hz_, at three time periods: Dawn (00:00–05:00), Dusk (18:00–23:00) and 06:00–17:00.

A second Pearson correlation between hourly mean SPL_500–2500Hz_ and indices was also performed on the entire acoustic dataset without considering the hour of the day as a grouping factor. Due to the fact that some of the noise affected the 500 to 2500 Hz frequency band, which was used as reference for chorusing activity, a complementary Pearson correlation was performed between indices and SPL obtained from the frequency band of 50 to 200 Hz. SPL_50–200 Hz_ was selected to be representative of the noise disturbance for both vessels and sediment transportation but not of fish calls. This step was intended to assess whether indices’ results were influenced by noise. In order to minimize the chance of including noise in SPL_500–2500Hz_ and to make this measure a better proxy of fish choruses, we used the SPL value only during the fish chorusing hours (i.e., 00:00–05:00 and 18:00–23:00) and used zero in the remnant hours of the day.

Furthermore, we verified the accuracy of the AIs and the C-H method by manually listening to the acoustic recordings for the fish choruses and by annotating the presence (1) or absence (0) of choruses. For this purpose, we chose 10 days (n = 240 h) of acoustic files for which we computed the response of the AIs and the C-H method. A confusion matrix [62] was built to summarize the performance of the indices based on the binary classification, and was formed by two dimensions: (1) fish chorus (presence/absence); (2) chorus detected by the acoustic index (yes/no). The threshold of detection was chosen to be >0.5 for ACI, ADI, BI and C, and <0.5 for H.

This classification created: (a) true positives (TP); i.e., the index detected the chorus when the chorus was actually present; (b) false positives (FP); i.e., the index detected the chorus when the chorus was not present; (c) true negatives (TN); i.e., the index did not detect the chorus when the chorus was not present; (d) false negatives (FN); i.e., the index did not detect the chorus when the chorus was present.

Based on those metrics, the accuracies and errors are given by Equations (5) and (6):(5)Accuracy = TP+TN/TP+FP+TN+FN

(6)Error rate = FN+FP/TP+FP+TN+FN

Finally, at location A1 and N1 for the monitoring duration mentioned in Table 1, the four-dimensional plot constructed represents hourly variation (*z*-axis) of H (*x*-axis) and C (*y*-axis), and the corresponding hourly mean SPL_500–2500Hz_ (fourth dimension) is represented as a color bar.

## 3. Results

### 3.1. General Soundscape Features

The underwater noise levels at monitoring regions A1 and N1 were dominated by fish chorusing at ~500 to 2500 Hz (Figure 2, labelled as X). Diverse noise inputs were recorded at A1 in 2017 (Figure 2b): pile driving at ~10 to 30 Hz (labelled as A); fishing boats passing in close proximity of the hydrophone (labelled as C) at ~10 to 150 Hz; and high waves and currents causing impact of sediment particles with the PAM recorder, resulting in noise at ~10 to 70 Hz (labelled as B). Sound levels at N1 were heavily influenced at ~50 to 200 Hz by continual vessel noise throughout the acoustic monitoring period (Figure 2c,d, labelled as Y). Table 3 shows mean (RMS), median and 95th percentile SPLs. Due to continual shipping activity, the low frequency (50–200 Hz) sound levels at N1 (95th percentile) were ~10 dB higher than in A1.

### 3.2. Temporal Pattern in Fish Chorusing

Figure 3 shows mean SPL (± SD) for the the 500 to 2500 Hz frequency band on an hourly basis at A1 and N1 during the spring period of years 2016 and 2017. Recurring nocturnal chorusing behavior, characterized by peaks of activity around dawn (01:00–05:00) and dusk (17:00–21:00), was observed.

### 3.3. Fish Chorus Detection Performance of AIs and the C-H Method

Figure 4a,g show the spectrograms of 4 and 26 April 2016 together with the corresponding hourly mean SPL and AIs. Fish choruses were registered around dawn (01:00–05:00) on both days (milder on the 4 April, more prominent on the 26 April). Intense chorusing activity occurred around dusk (17:00–21:00). As a result, SPL_500–2500Hz_ peaked during these periods.

ACI produced spurious peaks from ~00:00 to 19:00 on 9 April (Figure 4b) and from ~09:00 to ~15:00 on 26 April (Figure 4h). ACI failed to detect the dusk and dawn choruses. In particular, this technique displayed a deep decrease at ~03:00 on the 26th, corresponding to the dawn chorus, and it showed low values corresponding to dusk choruses of both days. ACI showed decreasing values with noise at low frequencies (10–300 Hz), and higher values with fish calls when not grouped in dense choruses.

On 9 and 26 April, the ADI values (Figure 4c,i) exhibited peaks at the chorusing hours, similar to SPLs. Nevertheless, additional false peaks occurred at ~06:00 to 10:00 on the 9 and at ~07:00 on 26 April.

The BI detected dusk choruses peaking at ~20:00 on both the 9 and 26 April. It failed to detect the dawn chorus on 9 April, but produced a false peak at ~06:00. On 26 April, BI detected the dawn chorus ~03:00 but produced other peaks at ~07:00 (Figure 4d,j).

C and H produced peaks and dips, respectively, during the same time periods of the peaks in SPL. In particular, during the fish chorusing hours, C produced peaks similar to those of SPLs, mostly at dawn but also at dusk (Figure 4e,k), whereas H resulted in dips of values (Figure 4f,l).

To have complete comprehension of the results, we investigated the properties of the C-H method by applying it to four distinct cases characterized by noise occurrence (Figure 5), discussed as follows.

***Case 1***: Intense dawn and dusk chorusing and sediment-generated noise.

Figure 5a depicts the dawn and dusk chorusing (~113 dB) and noise due to sediment transport (Figure 5a, label A). C and H produced peaks and dips during the dawn and dusk chorusing hours (Figure 5e,i). Beyond the chorusing hours, H and C remained invariant to any other noise occurring in the recording. It can be observed on the C-H plane that, during the chorusing periods, H and C fall on the left-side of the plane (Figure 5m, label F), and the periods without chorusing are gathered on the right-side of the plane (Figure 5m, label N).

***Case 2***: Mild dawn and dusk chorusing with vessel and sediment-generated noise.

Figure 5b demonstrates the point about noise from vessel reaching higher frequencies and covering fish sounds when it is high amplitude (labelled as B and D) and from sediment transportation (labelled as C). The dawn chorusing from ~00:00 to 05:00 was detected by H and C from ~02:00 (Figure 5f,j). H and C scantly responded to the dusk chorusing and to vessel noise at ~17:00 (label D). H and C, however, did not respond to vessel noise (Label B) and high-intensity sediment noise at ~12:00 to ~14:00 (Label C), respectively. On the C-H plane, the C and H during dawn chorus hours (Figure 5n, label F) were separated from the hours when chorusing was absent (Figure 5n, label N).

***Case 3***: Mild dawn chorus with noise from ~18 vessels.

Figure 5c shows a 24-h recording with dawn chorusing between ~02:00 to 04:00 with ~18 vessel passages in the background. H and C responded to the chorusing (Figure 5g,k), and outside chorusing hours, remained relatively constant (H ≈ 0.85 and C ≈ 0.27). As a result, the chorusing hours on the C-H plane were isolated from the others (Figure 5o).

***Case 4***: Absence of chorusing with continual, sediment-generated noise.

Figure 5d shows the continual sediment noise. H remained high, at ~0.88, whereas C was low at ~0.2, throughout the 24 h (Figure 5h,l). On the C-H plane, H and C for all periods were accumulated on the right-side of the plane (Figure 5p).

Similarly, we tested the AIs on the four cases (Figure 6). The ACI dipped when the fish calls became dense (Figure 6e–g) and thus was not able to catch the main fish chorusing hours, increasing for some intense and intermittent sediment-generated noise (Figure 6f). The ADI was high during the fish chorusing (Figure 6i,j) and non-chorusing hours (Figure 6j,k), but lowered for intense noise from vessels and sediments (Figure 6i,j). The BI was not consistent with fish chorusing and increased with vessel and sediment noise (Figure 6m–o).

When extending the analysis to the entire dataset, the Pearson correlations between the normalized hourly means of the results from indices and SPLs gave various results (Appendix A). At both A1 and N1, ACI, ADI and BI produced inconsistent and lower r values, during and beyond the chorusing hours. The C positively correlated with both the dawn and dusk chorus with r > 0.9 (Appendix A) and H negatively correlated with chorusing, thereby yielding r < −0.9 (Appendix A).

Results from the second Pearson correlation between hourly means SPL_500–2500Hz_ and indices are shown in Table 3. A highly significant correlation with SPL_500–2500Hz_ was found for H (−0.9 < r < −0.98; *p* ≈ 1) and C (0.89 < r < 0.95; *p* ≈ 0) for both sites and duration of monitoring (as in Table 2). ADI had correlation 0.61 < r < 0.81; *p* < 0.001, which was followed by BI’s (0.4 < r < 0.79; *p* < 0.001). A particularly weak linear relationship was found with ACI (−0.31 < r < −0.11). Weak or null correlations were found between hourly means SPL_50–200Hz_ and indices, with higher r presented by ADI in N1 (r = −0.41; *p* ≈ 0.19). In particular, C and H showed an r close to zero.

The confusion matrix created from the manual annotation of fish choruses in 240 acoustic files revealed high accuracy rates of H and C (95% and 83%, respectively; Figure 7j,k). The other AIs registered lower levels of accuracy (ACI: 60%, ADI: 59% and BI: 73%, respectively) and higher error rates (Figure 7g,i).

The 4-dimensional plots (Figure 8) show a distinct pattern during the dawn and dusk chorusing periods. Recording hours involving fish chorusing have distinct values of C and H (Label DC1 and DC2). The fish choruses are represented by low entropy and high complexity. Contrarily, the rest of the recordings, with or without noise presence, are represented by constant values of C and H (Label N), highlighting the robustness of C and H towards noise. Additional graphs representing fish chorusing trends obtained from H and SPL_500–2500Hz_ on the entire dataset are shown on 3-D contour plots on Appendix A.

## 4. Discussion

In this study, we introduced a technique based on the C-H method for detecting fish choruses and compared its detection performance with AIs, such as ACI, ADI and BI. It was shown that the chosen AIs were not consistent for detecting the fish chorusing, instead producing spurious detections, during chorusing hours. AIs resulted in low detection accuracy rates, whereas the introduced, C-H method showed accuracies of 95% and 83%. Moreover, H and C were significantly correlated with the fish chorusing. The C was found to be positively correlated with the fish chorusing, whereas H was negatively correlated, thus resulting in |r| > 0.9.

In this study, ACI decreased in the presence of dense fish chorusing activity (Figure 6e at ~02:00 and ~20:00; Figure 6f at ~04:00; Figure 6g at ~03:00). Nevertheless, ACI increased when there was mild chorusing activity (Figure 6f at ~00:00–01:00 and ~18:00–19:00; Figure 6g at ~00:00–01:00). Hence, ACI was able to detect the chorusing by increasing values when the fish started to aggregate songs but, as the chorus intensified, ACI tended to severely decrease. Studies in [23,33] noticed similar behavior of ACI, leading to the consideration that it fails to detect situations where calls or songs are so dense and undiscernible that they are interpreted by the index as a continuous sound. Indeed, ACI was designed to detect variation in intensities over time (milliseconds) but not constant sounds, such as vessel noise or compact fish choruses. ACI was found to decrease for the two loud vessel transits (Figure 6f at ~08:00 and ~17:00) and to relatively remain invariant (ACI ≈ 3800) for the transit of ~18 vessels in the background (Figure 6g). Contrarily, a study in [26] found that ACI increased for vessel noise. Further, ACI tended to decrease for mild sediment noise of short durations (~15 min) (Figure 6e at ~09:00; Figure 6h at ~18:00). When the sediment noise became louder and longer, however, ACI was not able to filter it (Figure 6f at ~12:00–14:00; Figure 6h at ~05:00–07:00). Studies have utilized amplitude [26] and frequency filters [26,56] for improving the detection capability of ACI, but such refining was sometimes insufficient to catch the peak of the chorus faithfully [21].

ADI increased for the dawn and dusk fish chorusing activity (Figure 6i–k). Accordingly, ADI showed r ≥ 0.6 during the chorusing hours (Table 4). ADI remained high for vessel transits (Figure 6j at ~17:00; Figure 6k). The study in [26] also demonstrated that ADI was influenced by vessel noise and responded to some sediment noise by producing peaks (Figure 6i at ~12:00–13:00 and at ~10:00–18:00). It is evident from Figure 6l that, for sediment noise occurring throughout the 24-h recording period, ADI exhibited random low and high values.

BI remained low for mild intensity chorusing and tended to increase as the chorusing intensity increased and became denser (Figure 6m at 00:00–04:00; Figure 6n at 00:00–06:00). BI was not able to detect the mild chorus at dusk (Figure 6n) or dawn (Figure 6o). BI remained low for vessel transits (Figure 6n at 16:00–18:00; Figure 6o) but responded to sediment noise (Figure 6n at 11:00–14:00; Figure 6p).

Long-term analysis of data from AIs failed to detect chorusing periods effectively and produced spurious detections beyond chorusing hours (Appendix A). AIs work on the principle of evaluating energy and its variations in the recordings measured in terms of power spectrum and Shannon’s entropy in the specified frequency range or over a selected amplitude threshold [20,61,63]. Nevertheless, when dealing with fish songs and long-term recordings, selecting these parameters can be elusive because it is likely that anthropophony will occur in the specified frequency range (as in Figure 5). Recently, a study [64] utilized an unsupervised clustering approach for detection of fish chorusing in a quiet environment. Those unsupervised techniques require prior knowledge of the fish calls. This may not be a reliable method for determining long-term fish chorusing trends in a natural habitat, where many fish species still remain unidentified.

The C-H method tested in this study was found to be efficient in addressing fish chorusing even in the presence of anthrophonies and geophonies. Consequently, it proved to be a useful tool to extract fish chorusing from acoustic recordings. This method does not require a specified amplitude threshold and frequency range. Rather, it works on the principle of extracting the information by attempting to quantify the randomness and correlations present in the acoustic recordings. Because, H is negatively correlated with the chorusing, it produces a dip during the chorusing hours. Contrarily, C is positively correlated with the fish chorusing; hence, it peaks during the chorusing hours, resembling the peaks in SPL (Figure 4, Figure 5, Figure 6 and Figure 7). The C-H method requires one to choose the embedding dimension 3 ≤ *d* ≤ 7, and the selection of d is simple and flexible. It is shown that C and H remains robust to varying *d* (see Appendix A), and it is recommended to utilize higher *d* = 6, 7 to retrieve more information, such as correlation structure and randomness from the acoustic recordings [50].

On the C-H plane (Figure 5m), the white noise located on the extreme right of the plane suggests that it is highly random (high H ≈ 0.93) and has no correlation structure (low C ≈ 0.13). The sine series is located on the extreme left of the plane, exhibiting determinism (low H ≈ 0.34) and increased correlation structure (high C ≈ 0.31). In the time period during the absence of fish chorusing, C and H lie on the right side of the plane. Similar to the position of white noise on the plane, as the fish chorusing starts, C and H tend to move left of the plane (Figure 5m–p). When the chorusing intensifies, C and H move close to the position of the sine signal (Figure 5m–p). This behavior occurs because fish chorusing will increase the correlation structure (C increases) and decrease the randomness (H decreases) in the background noise.

As the intensity and the duration of the chorus diminish, H (randomness) increases and C (correlation structure) tends to decrease. As a result, H and C move towards the right side of the plane (Figure 5m–p). In the absence of the fish chorusing, H and C remain relatively constant (Figure 5h,l,p). For instance, in a 24-h recording, where ~18 vessels transited and the dawn chorus occurred, the C-H method was robust to vessel passage and was able to detect the dawn chorus effectively (Figure 5c,g,k,o). On another day, C-H slightly responded to the vessel transit, which was heard for ~2 h (Figure 5f,j). This increase occurred because the long-duration vessel transit is composed of continual tones, which cause C in the acoustic time series to increase and H to reduce.

Our long-term data collected is well suited for understanding these AIs in terms of the robustness to different anthrophony and geophony. Several sources of anthrophony exist in the long-term recordings, including noises from various short-duration industrial and construction activities, seismic survey air gun blasting, noise from fishing vessels with various types of engines and long-duration noise from constant distant shipping. Together, that all forms an ideal complex marine soundscape to evaluate the performance of AIs in detection of fish choruses.

The establishment of efficient AIs for studying the variation of the dawn and dusk fish-chorusing trend will enable better understanding of species-specific ecology and behavior, based on acoustically derived information, such as abundance of calls, temporal occurrence and seasonal variability. The validation of ad hoc indices for automatic detection of biophony in marine environments could further facilitate the evaluation of the impact of anthropogenic disturbances [21,33]. This study shows the importance of application of effective AIs in establishing long-term biophony trends.

## 5. Conclusions

Marine acoustic biological activity could function as a highly suitable proxy for addressing trends in biodiversity levels and the functionalities of the ecosystems. In order to do so, adapted automatic information procedures able to discern biological sounds from other sources (anthropogenic or geophonic) are, today, in strong demand. The C-H method was herein tested in marine environments and intended to fill the gaps left by other indices that were ideated and firstly used for terrestrial sites. The C-H method was found to be strongly correlated with fish chorusing and not influenced by noise originating from shipping activity or natural sources, such as wind and tides. Future studies could apply C-H to terrestrial or freshwater recordings, where it could potentially be of help, particularly in those environments characterized by periodic overlapping choruses (such as bird dawn choruses) or where sounds produce high saturation of the spectrograms. The C-H method may also be extended for the detection of cetacean vocalizations, but other research is needed to verify this approach on different targets. Placed beside other existing acoustic indices, the C-H method could represent a complementary measure to determine changes in the composition of animal communities and help managers and decision-makers monitor wildlife.

## Figures and Tables

**Figure 1 entropy-21-00977-f001:**
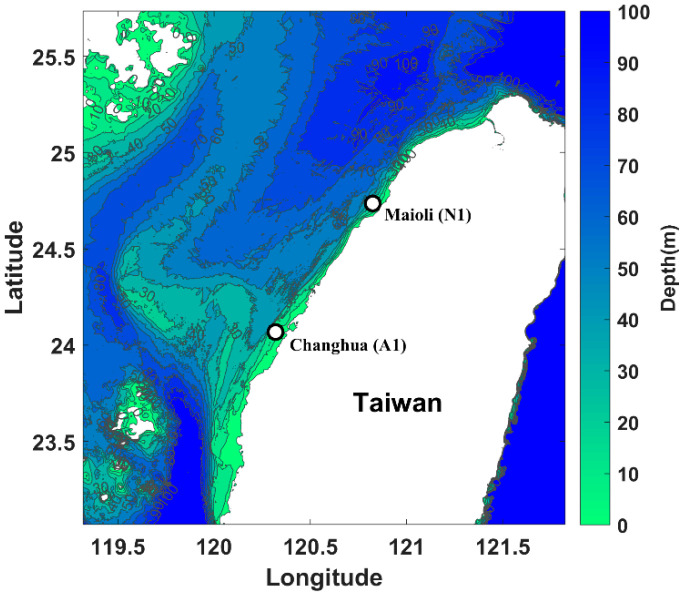
Location of the deployed passive acoustic monitoring (PAM) recording units at Changhua (A1) and Miaoli (N1) in the Taiwan Strait.

**Figure 2 entropy-21-00977-f002:**
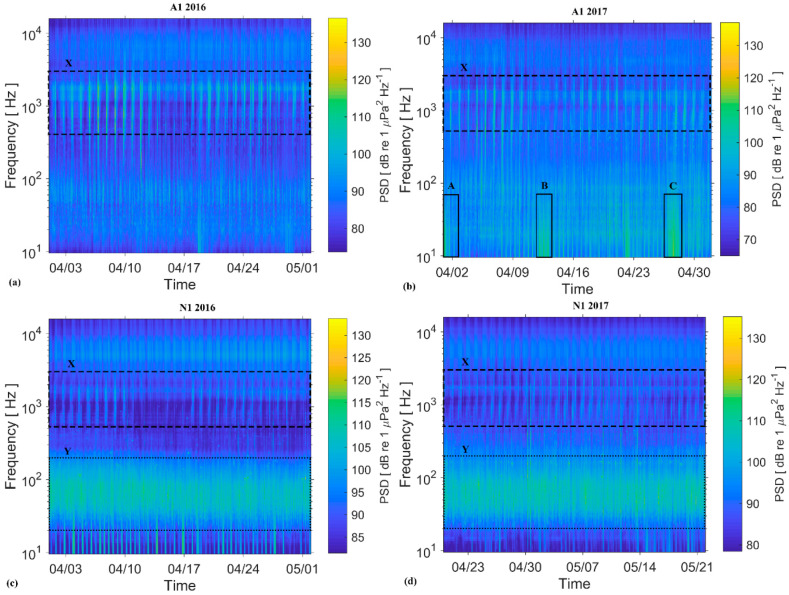
A 30-day spectrogram for different monitoring periods during 2016 and 2017 at locations A1 and N1. The color bar represents the sound levels at a given time and frequency. Label X: fish chorus; Y: continual shipping noise; A: pile driving drilling noise; B: sediment transport noise; C: fishing vessel passing in the vicinity of the recorder.

**Figure 3 entropy-21-00977-f003:**
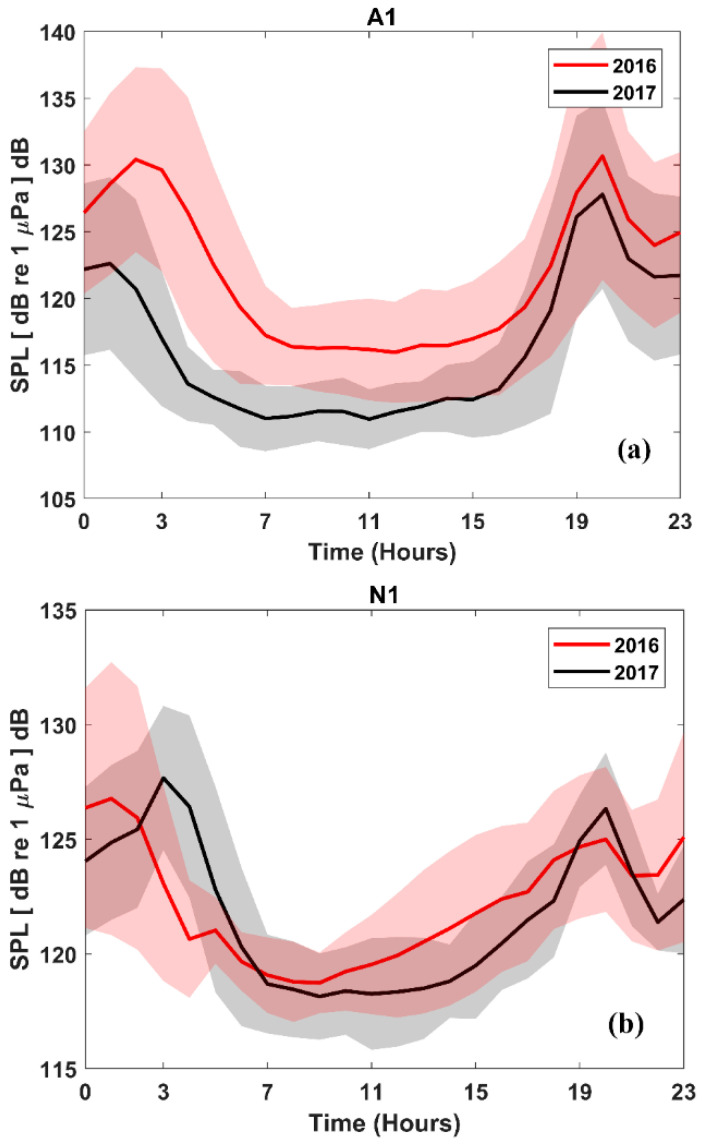
Sound pressure level (hourly mean and standard deviation) in the frequency band 500–2500 Hz at locations A1 and N1. Peaks around dawn and dusk are caused by fish chorusing behavior.

**Figure 4 entropy-21-00977-f004:**
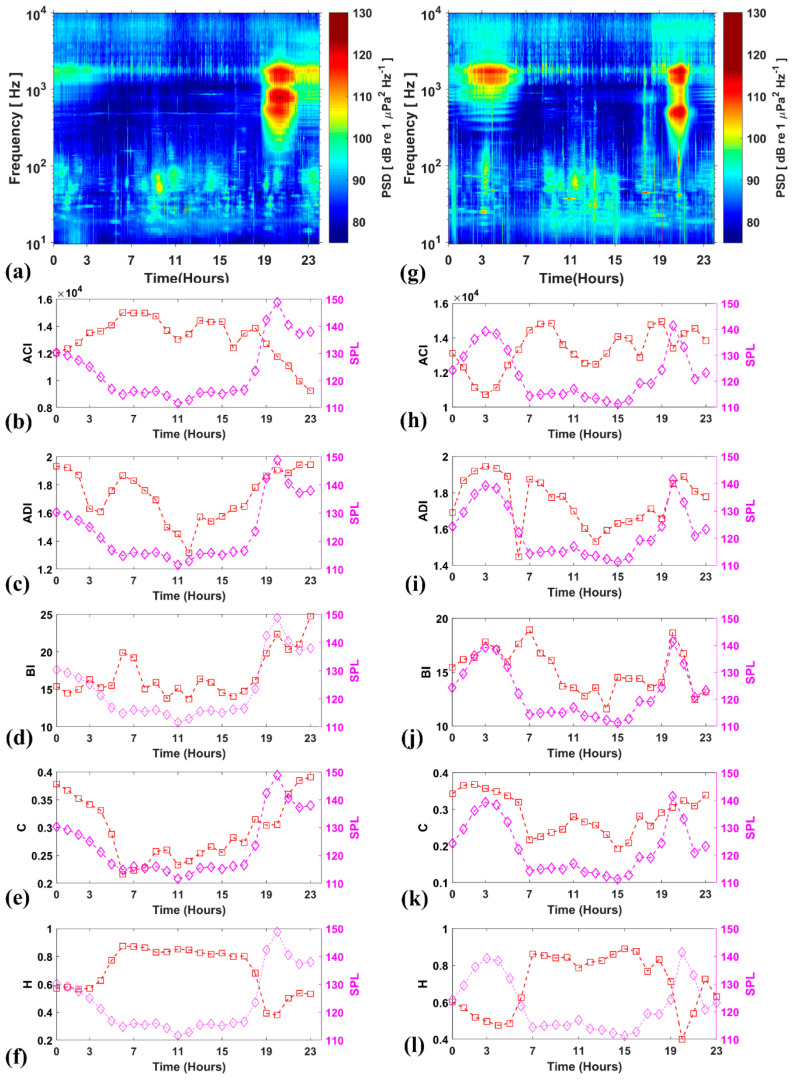
(**a**,**g**) A 24-h spectrogram of 9 and 26 April 2016, respectively at location A1. Hourly mean SPL_500–2500Hz_ (SPL, sound pressure levels). Hourly acoustic complexity index (ACI), acoustic diversity index (ADI), bioacoustics index (BI), permutation entropy (H) and permutation statistical complexity (C) for 9 April 2016 (**b**–**f**) and for 26 April 2016 (**h**–**l**).

**Figure 5 entropy-21-00977-f005:**
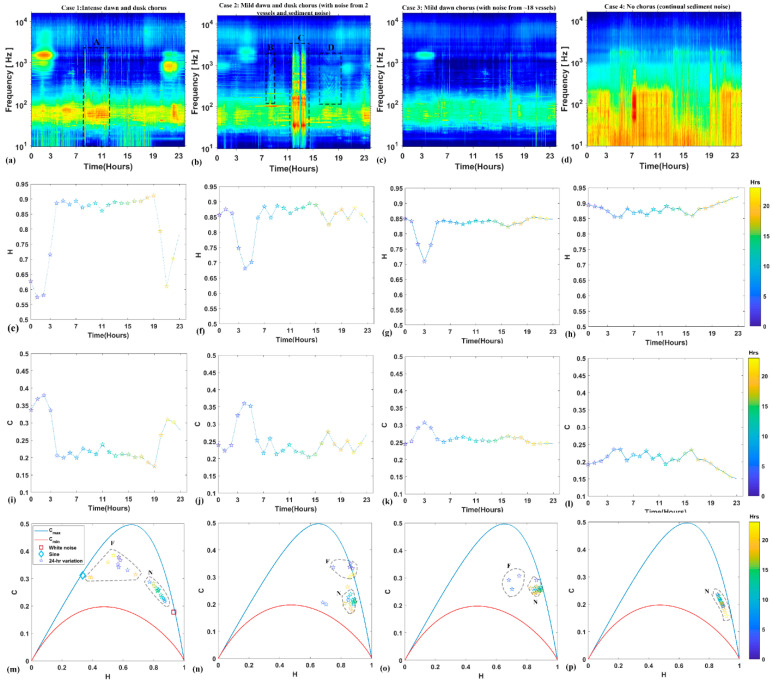
Graphs show a selection of interesting cases characterized by different levels of noise and chorusing activity. (**a**–**d**) 24-h spectrograms on four different days. Labels A and C indicate sediment transport noise. Labels B and D indicate vessel noise. Hourly H (**e**–**h**) and C (**i**–**l**) for the corresponding spectrogram are shown in (**a**–**d**). Hourly variations of C and H are plotted on the C-H plane (**m**–**p**), showing a separation of fish chorusing hours (F) from periods with no fish chorusing (N).

**Figure 6 entropy-21-00977-f006:**
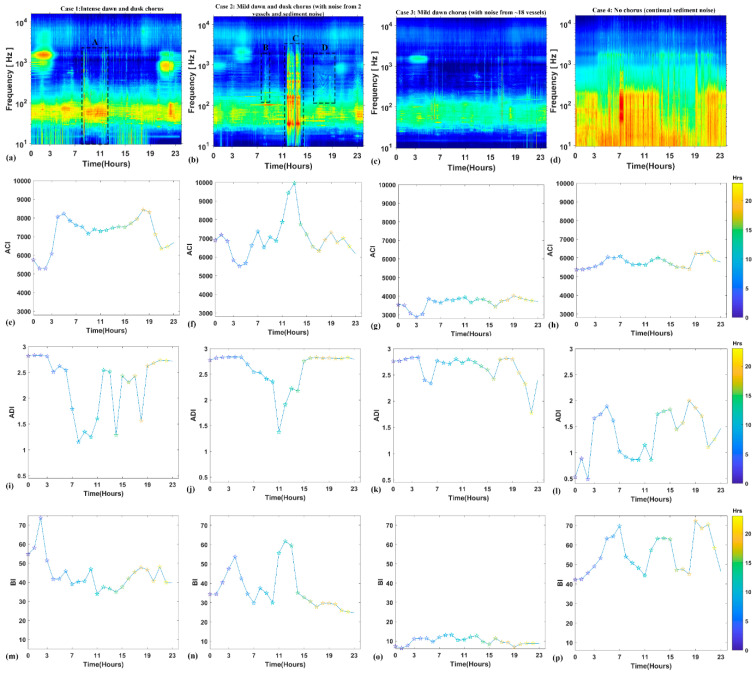
Graphs show a selection of interesting cases characterized by different levels of noise and chorusing activity. Fish chorusing are not clearly distinguishable on the bases of ACI, ADI and BI results. (**a**–**d**) 24-h spectrograms of the four different days. A,C: sediment transportation noise; B, D: vessel noise. Hourly ACI (**e**–**h**), ADI (**i**–**l**) and BI (**m**–**p**) evaluated for the corresponding spectrogram in (**a**–**d**).

**Figure 7 entropy-21-00977-f007:**
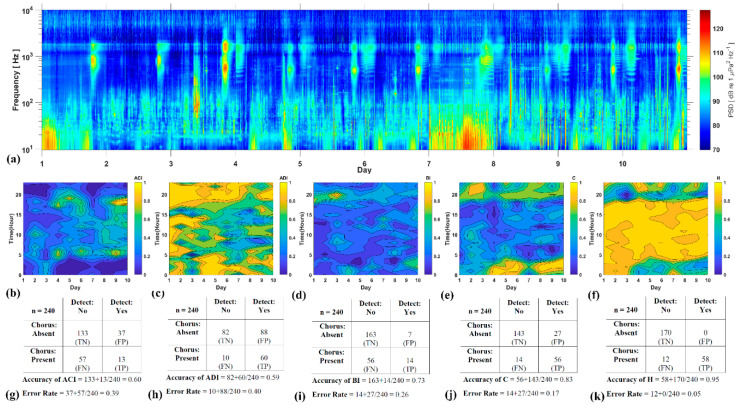
(**a**) Spectrogram representing 10 recording days (color bar represents the sound levels at a given time and frequency) and (**b**–**f**) 3-D contour plots showing AIs and C-H relative results (color bar represents the normalized indices value at a given day and time, 1-h resolution). (**g**–**k**) Confusion matrixes with accuracies and error rates for the corresponding indices.

**Figure 8 entropy-21-00977-f008:**
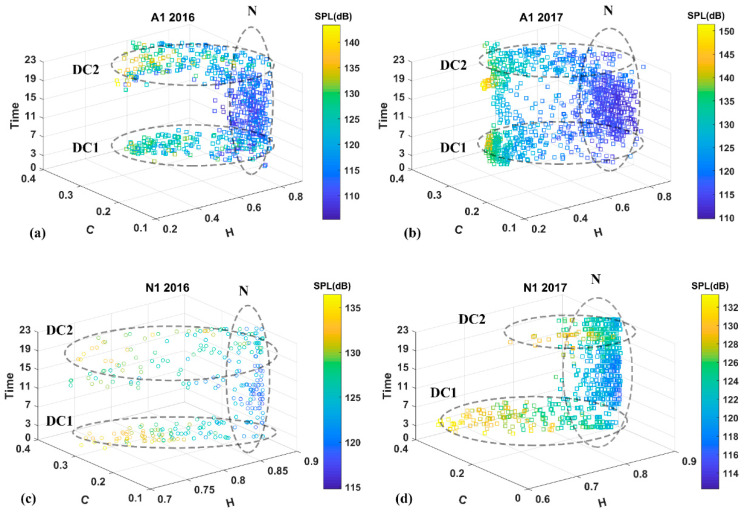
Four-dimensional plot representing the distribution of acoustic files on the bases of H, C, time in hours (x, y, z) and SPL (dB re: 1 µPa). (represented by the color bar). DC1: dawn chorus; DC2: dusk chorus; N: periods with no fish chorusing.

**Table 1 entropy-21-00977-t001:** Locations A1 and N1, and monitoring period and depth of the PAM recorders used in this study.

	Year	Month	Depth (m)	Latitude	Longitude
**A1**	2016	14 March–16 May	18.9	24°4.283′ N	120°19.102′ E
	2017	1 April–11 May			
**N1**	2016	1 April–7 May	19.3	24°44.080′ N	120°49.400′ E
	2017	20 April–24 May			

**Table 2 entropy-21-00977-t002:** Confusion matrix and metrics for binary classification.

	Chorus Detected: No	Chorus Detected: Yes
**Chorus:** **Absent**	TN	FP
**Chorus:** **Present**	FN	TP

**Table 3 entropy-21-00977-t003:** Mean RMS, median and 95th percentile of noise levels at monitoring regions A1 and N1 (dB re: 1 µPa).

Region	Frequency Band	YEAR	Mean RMS Level	MEDIAN	95th PERCENTILE
**A1**	**50–200 Hz**	2016	111.52	112.67	119.45
		2017	113.78	113.54	121.89
	**500–2500 Hz**	2016	121.88	120.45	138.23
		2017	116.42	113.92	132.00
**N1**	**50–200 Hz**	2016	127.29	129.10	133.21
		2017	128.23	128.89	133.56
	**500–2500 Hz**	2016	122.26	121.43	132.83
		2017	121.72	121.15	130.21

**Table 4 entropy-21-00977-t004:** Pearson’s linear correlation coefficient r (measure of the goodness-of-fit of the linear regression) and *p*-values (significance test for linear regression) evaluated between indices and SPL at two frequency bands 50–200 Hz (anthropophony and geophony) and 500–2500 Hz (biophony from fish chorusing).

Indices	A1				N1			
	2016		2017		2016		2017	
	50–200 Hz	500–2500 Hz	50–200 Hz	500–2500 Hz	50–200 Hz	500–2500 Hz	50–200 Hz	500–2500 Hz
**H**	r = −0.13	r = −0.98	r = −0.01	r = −0.97	r = 0.22	r = −0.93	r = 0.18	r = −0.9
	*p* = 0.73	*p* = 1	*p* = 0.53	*p* = 1	*p* = 0.08	*p* = 1	*p* = 0.59	*p* = 1
**C**	r = 0.14	r = 0.95	r = 0.18	r = 0.94	r = −0.19	r = 0.89	r = −0.16	r = 0.92
	*p* = 0.23	*p* = 0	*p* = 0.19	*p* = 0	*p* = 0.35	*p* = 0	*p* = 0.42	*p* = 0
**ACI**	r = −0.27	r = −0.31	r = 0.01	r = −0.11	r = −0.1	r = −0.3	r = 0.29	r = −0.2
	*p* = 0.9	*p* = 0.92	*p* = 0.48	*p* = 0.67	*p* = 0.012	*p* = 0.7	*p* = 0.7	*p* = 0.18
**ADI**	r = 0.09	r = 0.76	r = 0.17	r = 0.81	r = −0.41	r = 0.67	r = −0.21	r = 0.61
	*p* = 0.48	*p* = 0	*p* = 0.21	*p* = 0	*p* = 0.19	*p* = 0	*p* = 0.92	*p* = 0
**BI**	r = 0.16	r = 0.73	r = 0.21	r = 0.79	r = −0.11	r = 0.4	r = −0.2	r = 0.2
	*p* = 0.22	*p* = 0	*p* = 0.15	*p* = 0	*p* = 0.8	*p* = 0	*p* = 0.08	*p* = 0.12

## Data Availability

The authors declare that the data supporting the findings of this study are available within the article and its Appendix A, or are available from the corresponding authors upon request.

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
