# Peer review of "A Complexity-Entropy Based Approach for the Detection of Fish Choruses"

_entropy, 2019, doi:10.3390/e21100977_

Round 1

Reviewer 1 Report

This article describe the use of an audio feature, called C-H, in the context of fish population assessment. The audio feature is a derivative from the Shannon's entropy. Experiments show comparisons between C-H and a set of Acoustic Indices.

Overall, the paper is well written. The authors illustrate in details the computation of C-H. The choice of acoustic indices seem relevant.

However, the fish population, as well as presence of noises, is assessed only though the computation of hourly mean SPL on a precise frequency range.

From my viewpoint, the hourly mean SPL is another audio feature, that can be seen as an acoustic index. Hence, the different Ai and C-H are mostly tested from another acoustic index. It seem to me that the study could be improved through the use of other references for sound annotation (for instance human annotation) and/or a precise subsection that motivate the choice of mean SPL. This also should be discussed.

The test of C-H with synthetic sound is an interesting part because it is objective. I think this part should be extended (and placed in the experiment section).

Considering Figures, many plots are provided by the authors. The results are interesting, but it seem to me the multiplication of the Figure lose the reader rather than it gives a clear complementary viewpoint. The authors should consider to focus on a restraint set of Figures, or to place some of them in the supplementary material.

-----

Corrections:

114. "In this instance... uniform distribution" -> not clear

123. The statistical complexity measure was previously (line 76) referred to reference 51 (and not 49).

297. dips and peaks and inverted. See also line 305. The authors should keep the same order for all the article.

324. 8:00 - 11:00. This does not correspond to the time of the Figure.

354. r is not strictly above 0.9 at all times. This is also used in the discussion, but it seems wrong.

Author Response

Reviewer #1

This article describe the use of an audio feature, called C-H, in the context of fish population assessment. The audio feature is a derivative from the Shannon's entropy. Experiments show comparisons between C-H and a set of Acoustic Indices.

Overall, the paper is well written. The authors illustrate in details the computation of C-H. The choice of acoustic indices seem relevant.

We thank reviewer for this positive review.

However, the fish population, as well as presence of noises, is assessed only though the computation of hourly mean SPL on a precise frequency range.

From my viewpoint, the hourly mean SPL is another audio feature, that can be seen as an acoustic index. Hence, the different Ai and C-H are mostly tested from another acoustic index. It seem to me that the study could be improved through the use of other references for sound annotation (for instance human annotation) and/or a precise subsection that motivate the choice of mean SPL. This also should be discussed.

We deeply thank the referee for this comment that we think highly contributed to improve our paper. Following his/her suggestion, we used human annotation as an alternative way to evaluate the accuracy and performance of the AIs.

To carry-out this analyses, we have chosen 10 days of acoustic recording (240 hours) and have manually annotated the presence of fish chorus by assigning binary values for the presence (1) and absence (0) of chorusing. We provide full description in the manuscript at L248-254. A confusion matrix was utilized to solve the binary classification problem. By setting a threshold for the response of the AIs and CH method, the confusing matrixes (at L255-267) were derived and accuracy and error rate were evaluated. Please refer to newly added Figure 7 (At L390) and Table2 (L266), in the manuscript.

We thank the reviewer for this very important suggestion. This helped us to show empirically that the introduced CH method gives 95% (H) and 83% (C) accuracy for the detection of choruses.

The test of C-H with synthetic sound is an interesting part because it is objective. I think this part should be extended (and placed in the experiment section).

We thank the author for this comment; unfortunately, we think there was a misunderstanding. We did not use any synthetic sound, all sounds used in graphs and statistics derived from the field experiment dataset. We think that the referee is commenting on Figures 5 and 6. It is our opinion that to move this part in another section would create problems in the writing of the results and discussion. We apologize with the referee, but we would prefer to maintain the paper in this form, particularly now that we have added the new analyses and figure. We hope the referee can understand our point of view, otherwise we will find a way of matching his/her expectations.

Considering Figures, many plots are provided by the authors. The results are interesting, but it seem to me the multiplication of the Figure lose the reader rather than it gives a clear complementary viewpoint. The authors should consider to focus on a restraint set of Figures, or to place some of them in the supplementary material

We agree with the reviewer. We have moved the Figure 7 and 8 (In the previous version of the manuscript) to the supplementary information, now renamed as Figure S4 and S5.

Corrections:

"In this instance... uniform distribution" -> not clear

This means “when Shannon measure S[P]=0 then we can predict the outcome with complete certainty and in this instance, our knowledge of the underlying process described by the probability distribution P is maximum, whereas it is minimal if P is uniform distribution”. We have modified the sentence in order to rend it clearer and we have cited a suitable reference at L129.

The statistical complexity measure was previously (line 76) referred to reference 51 (and not 49).

Initially Lamberti et al. (Ref 52) introduced the information measure complexity for quantifying the distinguish degree of periodicity and dynamical features. Later Rosso et al. (Ref 41) combined the concept of complexity and permutation entropy to dynamically distinguish chaotic, periodic and stochastics dynamics.

At L77, when discussing the use of C-H method in our study we have cited Rosso et al., while introducing the complexity in the methods description we have cited Lamberti et al (L134).

dips and peaks and inverted. See also line 305. The authors should keep the same order for all the article.

We thank reviewer for identifying this mistake. We have corrected it at L320 and 328.

8:00 - 11:00. This does not correspond to the time of the Figure.

We thank reviewer for identifying this mistake. We have corrected it at L339. According to Figure 5b, the sediment transport take place from 12:00 hrs to ~ 14:00 hrs.

r is not strictly above 0.9 at all times. This is also used in the discussion, but it seems wrong.

We agree with the reviewer. Before, we had calculated the moving correlation between SPL and AIs which resulted in inconsistent r. In the revised version, we have evaluated r at three time periods: Dawn (00:00-05:00 hrs), Dusk (18:00-23:00 hrs) and 06:00-17:00 hrs (L235-237). After the reviewer suggestions, we have moved the Figure 7 and 8 (in the previous version) to supplementary information (now named as Figure S4 and S5).

Reviewer 2 Report

The Manuscript entitled "Complexity-Entropy based approach for detection of fish choruses" is an interesting attempt to compare the detection performance of the most used Acoustic Indexes (AIs) and to provide new tools for detecting sounds in marine environment. The Complexity-Entropy method was tested on the data to improve the current ability to detect fish choruses, and is applied to overcome the typical drawbacks of the AIs based on the Shannon entropy approach.

The Reviewer suggests that the Manuscript is suitable for publication after minor revision (correction to minor issues and text editing).

The Manuscript is very rich, but well structured and written. The presence of subheadings allows the reader to easily follow the MS structure, both in the Methods and in the Results section.

The Introduction section is balanced and clear. I suggest to insert a brief paragraph delineating the main characteristics of the Scienidae fish choruses. In the hypotheses, the Authors could also state whether they expect the C-H method to be better than the other AIs, according to previous research.

The Methods are very detailed, but well structured.

Table 2 appears in the text before Table 1 [LL.96]. I suggest to change the Table order by renaming the actual Table 1 as Table 2, and the other consequently. Actual Table 1 [LL.104] is a result, while actual Table 2 [LL.261] is a method.

LL. 121-123 The sentence should be rewritten after the comma.

The results are very well presented. Figures supporting the Author's findings are clear and useful.

In Figure 6 [LL. 345], Figure 7 [LL. 358] and Figure 8 [LL. 363] I suggest to erase the green boundary and the formatting marks.

The Discussion and Conclusion section are linear.

Supplementary materials are provided. 

Author Response

Reviewer #2

The Manuscript is very rich, but well structured and written. The presence of subheadings allows the reader to easily follow the MS structure, both in the Methods and in the Results section.

We thank the reviewer for the positive feedback.

The Introduction section is balanced and clear. I suggest to insert a brief paragraph delineating the main characteristics of the Scienidae fish choruses. In the hypotheses, the Authors could also state whether they expect the C-H method to be better than the other AIs, according to previous research.

We are grateful to the referee for his/her comments and advice.

As suggested, we have added a brief paragraph regarding main characteristics of the Scienidae family present at the Eastern Taiwan Strait with suitable references (L 108-111). In the supplementary materials we have also added a spectrogram showing with detail their calls.

We have added the hypothesis that CH method may perform better than other AIs, because of CH method ability to quantify both correlation and randomness in the acoustic recordings (L 89-91). We have also cited suitable reference (Ref 41).

The Methods are very detailed, but well structured

We thank again the reviewer for the positive feedback.

Table 2 appears in the text before Table 1 [LL.96]. I suggest to change the Table order by renaming the actual Table 1 as Table 2, and the other consequently. Actual Table 1 [LL.104] is a result, while actual Table 2 [LL.261] is a method.

We apologize for the mistake and we corrected it following reviewer’s suggestion. See L98 (Table 1) and 279 (Table 3, in the revised version).

121-123 The sentence should be rewritten after the comma.

We have rewritten the sentence at L 134-135.

The results are very well presented. Figures supporting the Author's findings are clear and useful.

We thank reviewer for this positive feedback.

In Figure 6 [LL. 345], Figure 7 [LL. 358] and Figure 8 [LL. 363] I suggest to erase the green boundary and the formatting marks.

In the revised manuscript, we have replaced all these figure to remove the formatting error.

The Discussion and Conclusion section are linear.

Supplementary materials are provided.

We again thank the reviewer for this positive feedback.

Reviewer 3 Report

Very interesting paper that demonstrates the inadequacy of standard acoustic indices in describing marine soundscapes. Authors present an alternative analysis method that provides a reliable indicator of fish chorusing despite the presence of variable non-biological noises.

However there are some obscure points that need clarification from the Authors (see general comments).

Here the main points I found worth of consideration :

General comments:

In the description of the acoustic environment where the study has been conducted, the Authors don't mention the possible presence of dolphins. It is important to know if the area is frequented by dolphins by collecting information from stranding records and from any other source of information e.g. sightings, or reports from fishermen.

Even if the work is mainly addressed at low frequencies, it is known that dolphins frequently feed on fish schools at night and may produce broadband echolocation clicks with some energy going down to few kHz and even to lower frequencies, in addition they also use echolocation pulses to elicit the resonance of swimbladders.

It is important to add some sentences to say something about dolphins in the area and, in case they may frequent the area, some info about their possible detection.

Spectrograms show impulsive noise in the frequencies above 5 kHz. Are you sure there are no dolphins around ? Have you considered that computing long time duration spectrograms by averaging on short duration time frames decreases the energy of impulsive sounds that, if emitted by dolphins can be only few hundreds of microseconds long ?

Have you made any analysis on the upper band ? e.g. to see a possible correlation with fish choruses ? in case of dolphins in the area this could means feeding activities when fishes aggregate ....

Authors also cite as a source of noise the sediment transportation but don't provide any cue about how they can demonstrate this. The spectrograms show increased level at low frequencies attributed to sediment transportation. However, sediment transportation increases with current speed, and the water flowing on the hydrophones makes low frequency noise and also high frequency noise if some turbulence is generated by some part of the submerged equipment.

Once these points are explained (required for acceptance and for making the paper a very valuable paper), I'd suggest very minor corrections and integrations here and there:

line 32: scientific bodies, maybe replace with scientists

line 33: add "offshore and coastal" before "construction .....

lines 47 to 50: 

this is not completely true, not all AIs are reliably proportional to biophony richness and complexity, in some bands AIs can be fooled by not biological environmental noises.

please remake the statements

line 83: maybe better to write "fluctuating ambient ocean noise"

line 104, table 1: indicate the name of the software package used for noise measures.

line 191: 

ship noise may have very variable frequency-time structures, ranging from constant low frequency tonal sound generated mainly by engines, to high frequencies generated by the trasmission gears, and also impulsive patterned noise generated by the cavitation of propellers, in particular the blades are damaged or dirty.

I think Authors should mention the heterogeneity of ship noises and how the different types of ship noise is interpreted by AIs

line 215: ... sediment generated noise. This type of noise is mentioned here for the first time. Authors should introduce it as a possible contribute to ocean noise in the introduction.

Also Authors don't explain how to recognize that noise is generated by sediment transportation and not by increased current speed.

line 219: "represents" replace with "to represent" or "to show"

line 252: impact of sediment particles ..... again, not demonstrated !

Author Response

Reviewer #3

Very interesting paper that demonstrates the inadequacy of standard acoustic indices in describing marine soundscapes. Authors present an alternative analysis method that provides a reliable indicator of fish chorusing despite the presence of variable non-biological noises.

We thank the reviewer for the positive feedback on the manuscript.

However there are some obscure points that need clarification from the Authors (see general comments).

Here the main points I found worth of consideration :

General comments:

In the description of the acoustic environment where the study has been conducted, the Authors don't mention the possible presence of dolphins. It is important to know if the area is frequented by dolphins by collecting information from stranding records and from any other source of information e.g. sightings, or reports from fishermen.

Even if the work is mainly addressed at low frequencies, it is known that dolphins frequently feed on fish schools at night and may produce broadband echolocation clicks with some energy going down to few kHz and even to lower frequencies, in addition they also use echolocation pulses to elicit the resonance of swimbladders.

It is important to add some sentences to say something about dolphins in the area and, in case they may frequent the area, some info about their possible detection.

We definitely agree with the referee. Dolphins/cetaceans do represent target species that absolutely deserves to be investigated. They are keystone species for the study of the marine ecology and its conservation and monitoring. The two stations chosen are located in Eastern Taiwan Strait (ETS). The ETS is known to have endangered indo-pacific humpback dolphins and bottlenose dolphins. As recommended by the referee, we have added this at L 111-113 with suitable reference (Ref 48,49). Unfortunately, in the recordings used in this study - during the spring season, we didn’t notice any significant clicks and whistles at frequency band we used.

Detection of dolphin vocalizations and echolocations in the long-term period in-between anthropogenic noise and snapping shrimps’ activity is one of the open problem existing in marine ecoacoutics and needs to be urgently solved. Unfortunately, detection of fish choruses happening for hours is relatively easier than compared to the dolphin vocalization, where whistles and clicks lasts few milliseconds. It is our firm opinion that the investigation of CH on dolphins vocalizations needs a standalone paper in order to be fully assessed and discussed. To address this issue, we are working on a second paper regarding the potential use of the CH method for dolphin vocalizations (on other set of recordings). However, we have added this hypothesis in the discussion section at L 510-512, together with the impellent need of further future research.

Spectrograms show impulsive noise in the frequencies above 5 kHz. Are you sure there are no dolphins around ? Have you considered that computing long time duration spectrograms by averaging on short duration time frames decreases the energy of impulsive sounds that, if emitted by dolphins can be only few hundreds of microseconds long ?

Have you made any analysis on the upper band ? e.g. to see a possible correlation with fish choruses ? in case of dolphins in the area this could means feeding activities when fishes aggregate ....

We thank reviewer for noticing the impulsive sounds at the frequency above 5 kHz. It is due to the presence of the snapping shrimps. However, the CH method was robust to the snapping shrimps.

For the moment, we did not perform any analysis on the high band, but focused our attention on fish calls. In order of not being misinterpreted, we clearly have stated it in all over the manuscript and in the title. However, we agree with the referee: since the premises of CH are very good, we are thrilled about proceeding to a successive study regarding CH test and verification on the upper frequency band.

Authors also cite as a source of noise the sediment transportation but don't provide any cue about how they can demonstrate this. The spectrograms show increased level at low frequencies attributed to sediment transportation. However, sediment transportation increases with current speed, and the water flowing on the hydrophones makes low frequency noise and also high frequency noise if some turbulence is generated by some part of the submerged equipment.

We understand the reviewer’s point of view and we have provided in the ms additional information regarding sediment transport and the relative noise (L102-108). The ETS experiences several tropical storms accompanied with strong wind speeds, which causes the rise in sea tidal level. This is the main reason for the sediment transport at the listening stations. The sediment transport at the ETS more often occur in summer (Typhoon season) and winter (Due to high wind speeds and increased sea tidal level).

We have empirically confirmed that the noise is generated by sediment transport by directly listening to the recordings and hearing particles/sediment hitting our recorded. Noise due to water flowing and waves was also present in those days, which are concomitant to harsh weather conditions declared by weather monitoring stations. However, noise generated by sediment transport was vastly the dominant source of geophony, and it was also the component of the soundscape more difficult to overcome by the indices.

Once these points are explained (required for acceptance and for making the paper a very valuable paper), I'd suggest very minor corrections and integrations here and there:

line 32: scientific bodies, maybe replace with scientists

As suggested, we have replaced this at L32.

line 33: add "offshore and coastal" before "construction ....

As suggested, we have replaced this at L33.

lines 47 to 50:

this is not completely true, not all AIs are reliably proportional to biophony richness and complexity, in some bands AIs can be fooled by not biological environmental noises.

We agree with the reviewer that existing AIs might respond to noise sources, as demonstrated in this study. Hence we have added the phrase ‘ideally AIs should be’ at L47.

please remake the statements

line 83: maybe better to write "fluctuating ambient ocean noise"

We have replaced this at L89

line 104, table 1: indicate the name of the software package used for noise measures.

We have replaced this at L214.

line 191:

ship noise may have very variable frequency-time structures, ranging from constant low frequency tonal sound generated mainly by engines, to high frequencies generated by the trasmission gears, and also impulsive patterned noise generated by the cavitation of propellers, in particular the blades are damaged or dirty.

I think Authors should mention the heterogeneity of ship noises and how the different types of ship noise are interpreted by AIs

We agree with the reviewer that the response of the AIs to heterogeneous shipping noise is data important to know. CH (or others AIs) are meant to be used in field recordings for long-term monitoring, thus all the evidenced noise sources are possible to be encountered. Understanding if some indices can work better in specific acoustic environments could make them useful in certain situations and not in others. Nevertheless, CH gave impressively good outcomes, particularly H accuracy levels, not particularly caring about the type of noise introduced.

We need to argument that Ecoacoustics indices (as the ones here presented) are never used to perform assessments on detailed sources. They address the community level, thus they use a broader resolution. This is why we feel the need to stress the fact that this paper focused on the detection of fish choruses versus all the remnant sounds. We apologize, but we could not have the possibility of going into details about all the singular shipping noise sources. For this question, it is our opinion that another standalone article would be needed (selecting diverse examples of noise that could be inserted in the same category and perform a statistics on the ability of the indices to detect them or not; and also considering the need to create categories on the base of the distance of sound source). Furthermore, to our knowledge it was never done before in such detail (i.e. considering the single noise sources and testing the results of diverse indices).

In order to better answer to this question, we have specified indices outcomes in presence of vessel noise in several different points of the paper. As mentioned in the manuscript the location N1 was located near the harbour (L107,108) which consists of various types of ships and boats transiting. We have utilized a long-term dataset of about 80 days of recordings and, as ecoacoustics demands, we gave broad results regarding the acoustic environment. Further, we have considered the cases of different vessels noise in Figure 5 and 6 to explain how AIs and CH method responded to it. We have also discussed the response of each index to the ship noise along all the discussion Section (L427-430 (ACI), 437(ADI), 444 (BI), 482 (CH)).

Nevertheless, we think that the issue raised by the reviewer is important and we will keep it mind as a valuable idea for a future work, which might address specifically diverse types of shipping noise and acoustic indices.

line 215: ... sediment generated noise. This type of noise is mentioned here for the first time. Authors should introduce it as a possible contribute to ocean noise in the introduction.

Also Authors don't explain how to recognize that noise is generated by sediment transportation and not by increased current speed.

Please see the answer at the previous point, in general comments, regarding sediment transportation. Here at L 215 (L226 in this version), the main reason of its selection was finding four different challenging cases to see how AIs and CH responded to different noise sources from vessels and sediment transport. We have confirmed the presence of sediment transport noise by direct aural and visual inspection of the recoding. The noise produced by hitting particles of sediment of the microphone was unmistakable. The 24-hour recordings in Figure 5a, 5b, 5c has the sediment transport, which were measured in summer (Figure 5a and 5b) and winter (5c). As said before, ETS experiences frequent storms and spurts in wind speed, which will trigger this sediment transport due to increased current speed and flooding.

line 219: "represents" replace with "to represent" or "to show"

As suggested, we have replaced this at L229.

line 252: impact of sediment particles ..... again, not demonstrated !

As previously said, we have confirmed this by listening to the actual recoding.

Round 2

Reviewer 2 Report

Reviewer#2 is appreciating the changes made to the MS, according to his/her previous revision. The reviewer also considered the changes made according to the suggestions of Reviewer#1. 

For this reasons, I suggest to accept the MS in its present form.